# RAG Enabled Conversations about Household Electricity Monitoring

Carolina Fortuna[1,*], Vid Hanžel[1] and Blaž Bertalanič[1]

[1]*Jozef Stefan Institute, Ljubljana, Slovenia*

## Abstract
In this paper, we investigate the integration of Retrieval Augmented Generation (RAG) with large language models (LLMs) such as ChatGPT, Gemini, and Llama to enhance the accuracy and specificity of responses to complex questions about electricity datasets. Recognizing the limitations of LLMs in generating precise and contextually relevant answers due to their dependency on the patterns in training data rather than factual understanding, we propose a solution that leverages a specialized electricity knowledge graph. This approach facilitates the retrieval of accurate, real-time data which is then synthesized with the generative capabilities of LLMs. Our findings illustrate that the RAG approach not only reduces the incidence of incorrect information typically generated by LLMs but also significantly improves the quality of the output by grounding responses in verifiable data. This paper details our methodology, presents a comparative analysis of responses with and without RAG, and discusses the implications of our findings for future applications of AI in specialized sectors like energy data analysis.

## Keywords
retrieval augmented generation, large language models, electricity knowledge graph, households

## 1. Introduction

Due to growing population and technological advances, global electricity consumption, and consequently also $CO_2$ emissions are increasing [1]. The residential sector makes up 25% of global electricity consumption and has great potential to increase efficiency and reduce $CO_2$ footprint without sacrificing comfort [2]. Stakeholders such as government and regulatory bodies, electricity system operators as well as individual household are increasingly relying on data for day to day operations and decision making [3, 4, 5]. With Large Language Models (LLMs) disrupting existing established work processes promising increased efficiency, we expect their adoption also in supporting such stakeholders.

However, in some cases, LLMs are not reliable because they can generate plausible-sounding but incorrect or nonsensical answers due to their reliance on patterns in training data rather than true understanding. They are prone to "hallucinating" information, where they produce details that are not based on factual data, and they can also struggle with ambiguous or context-dependent queries [6]. Additionally, their outputs can reflect biases present in their training data, leading to potentially harmful or biased responses [7]. Finally, their capabilities are enabled by their training data, encountering limitation with unseen private sources.

The limitations, including inherent uncertainty and variability make LLMs less dependable for applications requiring high accuracy and factual consistency such as data informed electricity related policy making and regulations. To mitigate for such cases, it is possible to 1) fine tune the LLMs on some specific data to adapt it to a particular task or domain [8], 2) adopt a two phase process involving context retrieval from external sources (such as databases or documents) in response to a query and uses this information to generate a more accurate and contextually relevant answer [9] or 3) prompt engineering by crafting and refining the inputs (prompts) given to a language model to elicit the best possible responses [10].

*Woodstock'22: Symposium on the irreproducible science, June 07–11, 2022, Woodstock, NY*

*Corresponding author.

✉ carolina.fortuna@ijs.si (C. Fortuna); vid.hanzel@gmail.com (V. Hanžel); Blaz.Bertalanic@ijs.si (B. Bertalanič)

🆔 0000-0003-0547-3520 (C. Fortuna); 0009-0007-9989-0136 (V. Hanžel); 0000-0001-9944-0732 (B. Bertalanič)

Each of the three methods is best suited for different scenarios. LLM fine-tuning is ideal for applications that demand high precision and customization in specific domains, such as specialized customer service or technical advice in fields like healthcare or finance [8]. Prompt engineering is particularly useful for quickly improving model outputs across various tasks without the need for further training, making it suitable for content creation or rapid prototyping [10]. Relying on external data sources, also referred to as retrieval augmented generation (RAG) is especially advantageous for tasks that require current and detailed information, such as answering factual questions or generating comprehensive reports, as it dynamically integrates the latest data into the model's responses [9].

Nevertheless, in spite of the promises of RAG, a very recent study regarding the reliability of legal tools that use it [11] found that even with RAG there may be hallucinations between 17% and 33% of the time.

The first studies exploring the capabilities of LLMs and chatbots in the electric energy sectors are coming out [12], emphasizing opportunities and identifying challenges such as scarcity of domain-specific data in the pre-training of LLMs. At the same time, more structured electricity datasets are also emerging [13]. However, to date, investigations on the potential enhancement of LLM responses via RAG using electrical energy data is missing.

This paper is the first to assess and report the RAG-based question answers related to household electrical energy measurement aspects. Relying on the recently published electricity consumption knowledge graph, we study the capabilities of ChatGPT, Gemini and Llama in answering electricity related questions. Furthermore, we compare the answers with the ones generated through a RAG techniques that leverages an existing electricity knowledge graph. We find that RAG helps with more precise responses most in most of the cases. In other cases, it ignores some of the factual information possibly due to the bias build in during training while in other cases it may provide irrelevant answers.

The paper is organized as follows. Section 2 provides background of RAG, Section 3 discusses KG based RAG for electricity while Section 3.1 illustrates the concrete approach employed for this study. Section 4 provides the analysis while Section 5 concludes the paper.

## 2. Background on Retrieval Augmented Generation

Retrieval Augmented Generation (RAG) is a technique that combines the capabilities of retrieval-based and generation-based models to improve information generation and accuracy. As depicted in Figure 1, in RAG, the LLM produces and answer based on three elements: a question, a query and context. The question is typically provided by a human in natural language. It is mapped by a transformation block to a query as depicted by steps 1 and 2 in Figure 1. The transformation step may embed the question and the embedding subsequently represents the query. It can also transform the question from natural language to an alternative query language such as SQL or SPARQL. The query resulting from the transformation block is then sent to a retrieval system that fetches relevant documents or pieces of information from a large dataset as represented by steps 3 and 4 in the figure. This system can use a vector database when the query is an embedding, and relational database in case of an SQL query or a knowledge graph in the case of a SPARQL query. The information retrieved by the system (i.e. embeddings, records, triples) is referred to as context that is send together with the question and query to the LLM that generates the answer as illustrated in steps 5 and 6. With this process, the generative model produces a coherent and contextually accurate response. This approach leverages the extensive factual knowledge in the retrieval database and the linguistic and contextual abilities of generative models.

In general, RAG is considered useful because it enhances the accuracy and reliability of generated content by grounding it in real, retrieved data, reducing the likelihood of generating hallucinations or incorrect information [9]. This makes RAG particularly valuable in applications requiring precise and factual responses, such as question answering, customer support, and content creation, where both relevance and correctness are critical. By combining retrieval and generation, RAG achieves a more robust and informed response generation process.

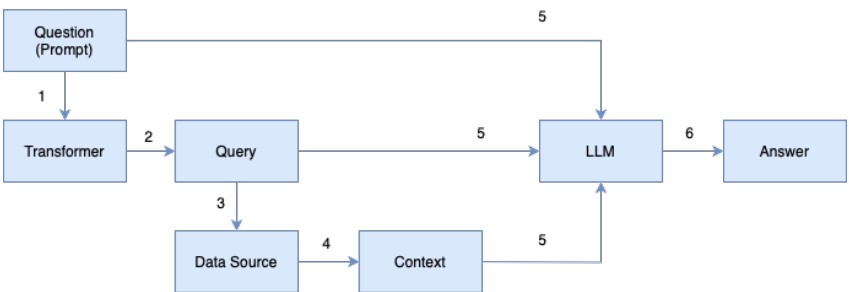

**Figure 1:** High level overview of the RAG process.

## 3. RAG with an Electricity Knowledge Graph

To better understand the potential of RAG in enabling more accurate conversation for energy stakeholders, we study a case using a recently proposed electricity knowledge graph [13] depicted in Figure 2. The KG is encoded in RDF, connected to Wikipedia and DBpedia, stored in Blazegraph and can be queries via SPARQL. So we assume that the transformation block from Figure 1 transforms the natural language to the corresponding SPARQL query, and that the LLM receives the natural language question, the SPARQL query and the triples that are retrieved from the KG. We evaluate the RAG with two versions of ChatGPT (4o and 4), Gemini 1.5 and Llama-3-8b-chat-hf and select questions that require increased precision as follows:

- *Prompt 1* "Enumerate in one short sentence the electricity consumption datasets collected in the UK." (For Gemini it returned nothing so we rephrased to: "How many electricity consumption datasets were collected in the UK?")
- *Prompt 2* "Enumerate in one short sentence the available electricity datasets located in countries with a GDP per capita higher than $50000."
- *Prompt 3* "Enumerate in one short sentence the available electricity datasets that are not located in Europe and are located in a place with a high education level."
- *Prompt 4* "Can you explain the load profile of house 1 in the REFIT dataset?"

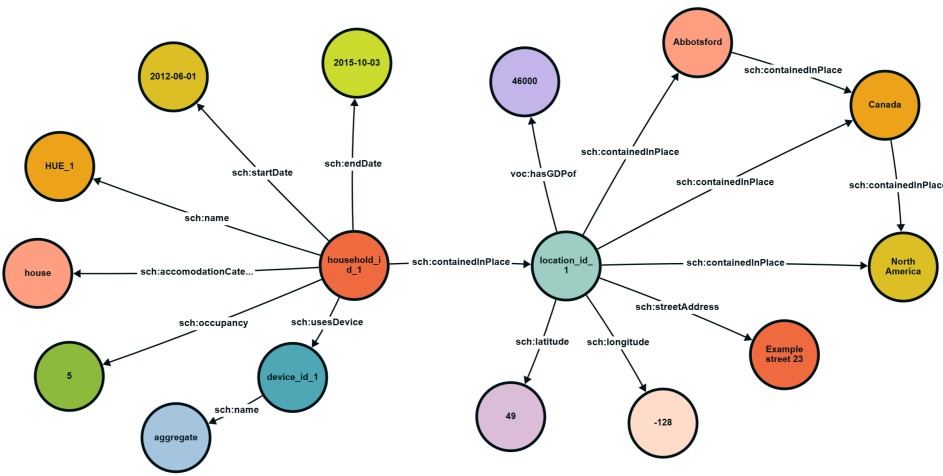

**Figure 2:** Electricity Knowledge graph visualization [13].

### 3.1. ChatGPT 4o and RAG Example

To better understand the methodology employed in this paper, we provide an example with ChatGPT4o. Given Prompt 1, we first use the regular non-RAG approach followed by the RAG approach as listed in

the boxes below. It can be seen that the response to the natural language question returns five datasets, i.e. NEED, ECUK, Smart Meter Energy Consumption Data, and a pointer to a portal. The response with the RAG returns the list of datasets available in the electricity KG. Only one of the datasets, namely REFIT, overlap between the two answers.

---

*Prompt 1* for ChatGPT 4o - no RAG

Enumerate in one short sentence the electricity consumption datasets collected in the UK?

---

Answer

In the UK, key electricity consumption datasets include the National Energy Efficiency Data-Framework (NEED), the annually updated Energy Consumption in the UK (ECUK) datasets, real-time Smart Meter Energy Consumption Data, regional and local Sub-National Electricity Consumption Data, detailed data from the Household Electricity Survey, diverse datasets from the National Grid Electricity System Operator (ESO) Data Portal, and the REFIT Smart Home Dataset.

---

*Prompt 1* for ChatGPT 4o with RAG

*Question*:
Enumerate in one short sentence the electricity consumption datasets collected in the UK?

*Query*:
PREFIX voc: <https://elkg.ijs.si/ontology/>
PREFIX saref: <https://saref.etsi.org/core/>
PREFIX rdf: <http://www.w3.org/1999/02/22-rdf-syntax-ns#>
PREFIX schema: <https://schema.org/>
SELECT DISTINCT ?prefix ?countryName WHERE {
?country rdf:type schema:Country .
?country schema:name ?countryName .
?house rdf:type schema:House .
?house schema:name ?houseName .
FILTER(?countryName = "United Kingdom") .
BIND(STRBEFORE(?houseName, "_" ) as ?prefix)
}

*Enhanced Context*:
prefix countryName
ECDUY United Kingdom
IDEAL United Kingdom
IAWE United Kingdom
REFIT United Kingdom
EEUD United Kingdom
ENERTALK United Kingdom
HEART United Kingdom
HUE United Kingdom
PRECON United Kingdom
SUST1 United Kingdom
UCIML United Kingdom
LERTA United Kingdom
DRED United Kingdom
UKDALE United Kingdom
DEDDIAG United Kingdom
DEKN United Kingdom
REDD United Kingdom
SUST2 United Kingdom
ECO United Kingdom
HES United Kingdom

**Answer**

The electricity consumption datasets collected in the UK include ECDUY, IDEAL, IAWE, REFIT, EEUD, ENERTALK, HEART, HUE, PRECON, SUST1, UCIML, LERTA, DRED, UKDALE, DEDDIAG, DEKN, REDD, SUST2, ECO, and HES.

## 4. LLM and RAG Analysis

In this section we summarize and analyze the responses provided by the selected LLMs to the four questions without employing RAG and with RAG.

### 4.1. Analysis related to Prompt 1

Ther answers to Prompt 1 are summarized in Table 1. It can be seen from the first row of the table that ChatGPT 4o provided 6 datasets and one data repository as answer without employing RAG and 20 datasets by employing RAG. There is one overlap in the found datasets with the two approaches. Overall, it provided the most precise answers given the natural language question. The answer of ChatGPT 4 was of equal quality, however expressed in more than as short sentence, than instructed. In the truncated answer in the box below it can be seen that ChatGPT 4 lists and expands the acronyms of the datasets and provides a little unsolicited explanation.

**Table 1**

Quality of the answers to Prompt 1: "Enumerate in one short sentence the electricity consumption datasets collected in the UK." (For Gemini it returned nothing so we rephrased to: "How many electricity consumption datasets were collected in the UK?").

| LLM | non-RAG Answer | RAG Answer | Overlap | Comments |
| --- | --- | --- | --- | --- |
| ChatGPT 4o | 6 datasets, 1 repo | 20 datasets | 1 dataset | |
| ChatGPT 4 | 6 datasets, 1 repo | 20 datasets | 1 dataset | Lengthier than solicited textual response. |
| Llama-3-8b-chat-hf | 0 datasets | 19 datasets | 0 datasets | Sources that collected data rather than datasets. |
| Gemini | 1 dataset and 2 repos | 5 datasets | 0 datasets | Gives examples and pointers, avoids being precise. |

**ChatGPT 4 RAG Truncated Answer to Prompt 1**

Here are the electricity consumption datasets collected in the UK, each with a unique prefix indicating the dataset source:
ECDUY: Electricity Consumption and Demand Usage Yearly
IDEAL: Individual Dwelling Electric Appliance Load
IAWE: Individual Appliance and Whole house Energy
....

These datasets encompass a variety of focuses, from detailed appliance-level monitoring to whole-house energy usage, supporting research in energy consumption patterns, efficiency analysis, and predictive modeling.

As can be seen in the 3rd row of Table 1, Llama-3-8b-chat-hf does not provide any datasets when not using RAG. Instead it provides sources that collected data as can be seen from the answer in the box below. When using RAG, Llama-3-8b-chat-hf lists 19 out of the 20 datasets provided as context. This may be due to build-in bias that disregards on the the respective datasets. Such behavior is consistent with some of the remarks in [11]. For the case of Gemini, in both non-RAG and RAG approaches, the chatbot avoids being specific in its answers. It however gives examples and references to where to search more as illustrated in the second and third boxes below.

> ### Llama-3-8b-chat-hf non-RAG Answer to Prompt 1
>
> The UK has collected electricity consumption datasets from various sources, including National Grid, BEIS, OpenEI, UK Power Networks, ONS, and research institutions, covering different time periods, geographic areas, and data granularity.

> ### Gemini non-RAG answer to Prompt 1
>
> There isn't a single definitive source that provides a count of all electricity consumption datasets collected in the UK. However, several resources indicate the existence of multiple datasets:
> The UKERC Energy Data Centre catalog lists datasets related to electricity consumption [1]. While it doesn't provide a total count, it highlights various datasets.
> A sample of anonymized electricity consumption data is available, implying there's a larger dataset used for national statistics [2].
> A dataset on Kaggle provides information on electricity demand in Great Britain, suggesting collection of electricity consumption data [3].
> These examples showcase multiple datasets on electricity consumption in the UK, but not a specific count.

> ### Gemini RAG answer to Prompt 1
>
> The UK has electricity consumption datasets collected through projects like ECDUY, IDEAL, IAWE, REFIT, EEUD, and others listed in the enhanced context.

## 4.2. Analysis related to Prompt 2

The answers to Prompt 2 are summarized in Table 2. It can be seen from the first row of the table that ChatGPT 4o provided 2 datasets, i.e. REFIT and ECO, without employing RAG and 5 datasets by employing RAG. There is one overlap in the found datasets with the two approaches. The answer of ChatGPT 4 was of 3 datasets, i.e. Pecan, REDD and GREEND without RAG and 5 datasets with RAG and with the REDD dataset overlapping. It can be seen that the answers of ChatGPT 4o and ChatGPT 4 vary and this variation is due to the fact that country GDPs have changed between the training time of the two models. The UK, where REFIT was collected, used to have below 50k $ GDP in the past while this has recently changed according to the IMF [1]. We can conclude that the answers with RAG are more predictable as it could retrieve examples with GDP at the time of collection, last year or now. However, when such tools are integrated into the standard enterprise business processes, safeguards that control for untimeely information should be considered.

**Table 2**
Quality of the answers to Prompt 2: "Enumerate in one short sentence the available electricity datasets located in countries with a GDP per capita higher than $50000."

| LLM | non-RAG Answer | RAG Answer | Overlap | Comments |
| --- | --- | --- | --- | --- |
| ChatGPT 4o | 2 datasets | 5 datasets | 1 dataset | REFIT, ECO |
| ChatGPT 4 | 3 datasets | 5 datasets | 1 dataset | Pecan, REDD and GREEND |
| Llama-3-8b-chat-hf | 0 datasets | 5 datasets | 0 datasets | Countries with 50k GDP and their agencies that collect data. |
| Gemini | 0 datasets | 0 datasets | 0 datasets | Gives examples and pointers, avoids being precise, finds problems in the SPARQL query. |

As can be seen in the third line of Table 2, Llama doesn't return any dataset in the non-RAG approach, however as can also be seen in the box below, it finds countries that respect the GDP constraint and provides the agencies in the respective countries that collect such data as can be seen in the box below.

---

[1]https://www.imf.org/external/datamapper/NGDPDPC@WEO/GBR/DEU/GRC

With RAG, the answer is consistent with the datasets provided as context. For this prompt, Gemini does not identify any datasets in either approach as can be seen from the last row of the table. It provides a generic answer in the non-RAG version and finds issues in the SPARQL query in the RAG version.

> **Llama-3-8b-chat-hf answer**
>
> Here are some available electricity datasets located in countries with a GDP per capita higher than $50,000:
> United States: Energy Information Administration (EIA) datasets on electricity generation, consumption, and prices
> Canada: Natural Resources Canada datasets on electricity generation, consumption, and prices
> .....
> Note that this is not an exhaustive list, and there may be other datasets available in these countries. Additionally, some datasets may be available through international organizations such as the International Energy Agency (IEA) or the Organization for Economic Cooperation and Development (OECD).

## 4.3. Analysis related to Prompt 3

The answers to Prompt 3 are summarized in Table 3. It can be noticed that the non-RAG version provides limited answers, even with the more capable ChatGPT versions in the first two rows. Furthermore, ChatGPT 4 refers to electricity pricing datasets rather than electricity datasets collected when the price was higher than 0.25€/kWh. ChatGPT 4o captured the nuance and returns one correct dataset that overlaps with on the the datasets returned using RAG. The non-RAG answers with Llama and Gemini are not useful for this prompt indicating relatively shallower semantic capabilities. The answers generated with the RAG versions seem to omit the information provided as context. Further research may be required to understand whether the more precise the question, the more it confuses the LLM that becomes unable to leverage the query and context in the response generation.

**Table 3**
Quality of the answers to Prompt 3: "Enumerate in one short sentence the available electricity datasets that are located in Europe and had at the time of recording an electricity price higher than 0.25€/kWh."

| LLM | non-RAG Answer | RAG Answer | Overlap | Comments |
| --- | --- | --- | --- | --- |
| ChatGPT 4o | 1 dataset | 10 datasets | 1 dataset | ECO |
| ChatGPT 4 | 2 datasets | 10 datasets | 0 dataset | Pricing datasets, not measurement. |
| Llama-3-8b-chat-hf | 0 datasets | 8 datasets | 0 datasets | List of national TSO having wholesale prices. |
| Gemini | 0 datasets | 7 datasets | 0 datasets | Non-RAG provides a statement that doesn't make sense. |

## 4.4. Analysis related to Prompt 4

The fourth prompt assesses the value of retrieving a load profile image from the KG for understanding the consumption behavior of that respective house. As expected, when asking ChatGPT 4o and 4 the following: "Can you explain the load profile of house 1 in the REFIT dataset?", it provides a well structured, but generic answer. It provides valuable guidelines on how to analyze the house 1 REFIT load profile as can be seen in the first box below. However, when adding the load profile image, extracted from the KG, both versions are able to provide specific analysis based on the numbers present in the image. However, as can be seen from the prompt and answer in the boxes below, ChatGPT 4 made less factual mistakes interpreting the image compared to ChatGPT 4o. ChatGPT 4o identifies a drop between 1-6 hour after after a peak from 0 to 1 hour that is not present in the load profile. ChatGPT 4 extracts the morning peak 0-5 hours followed by the morning drop 5-10 hours that is more consistent with the profile.

The REFIT dataset is a detailed collection of electrical consumption data from several households in the UK, gathered as part of the REFIT (Retrofit) project. The load profile of a house in this dataset refers to the pattern of electrical usage over a specified period. Specifically for house 1 in the REFIT dataset, the load profile can be analyzed by looking at the time-series data of electrical consumption recorded.

Here's a step-by-step explanation of what the load profile might reveal:

**Data Collection**

1. *Time Series Data*: The dataset contains time-stamped readings of electrical usage. Each entry typically records the power consumption (in watts or kilowatts) at regular intervals (e.g., every minute).

**Key Components of the Load Profile**

2. *Daily and Weekly Patterns*:

- *Daily Variations*: The load profile can show how electrical consumption varies throughout the day. Peaks might correspond to times when occupants are home and using appliances (morning and evening), while troughs could indicate periods of inactivity (late night).

- *Weekly Variations*: Similarly, the profile might reveal differences between weekdays and weekends, reflecting different routines and usage patterns.

3. *Seasonal Variations*:

- Changes in consumption due to seasonal factors, such as increased heating in winter or air conditioning in summer.

.....

**Example Interpretation**

For house 1, the load profile might show:

- Morning peaks as occupants prepare for the day (e.g., 7-9 AM).

- Evening peaks when occupants return home and use appliances (e.g., 6-9 PM).

- Lower consumption during working hours if the house is unoccupied.

.....

If you have access to the actual data, we could plot and analyze specific examples to provide more detailed insights.

## Question for ChatGPT 4o

Can you explain the load profile of house 1 in the REFIT dataset? (added load profile plot for context)

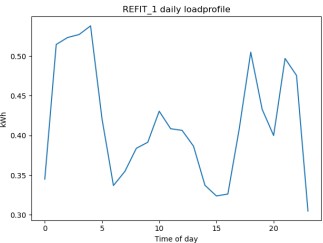

## Example portion of factual answer ChatGPT 4o vs 4

ChatGPT 4o

1. **Early Morning Peak (Around 0 to 1 hour)**:

- The load profile shows a significant peak in the early morning hours. This could be due to the use of appliances such as heating systems, or it might indicate the end of a high-consumption activity that started late in the previous evening.

2. **Drop After the Peak (Around 1 to 6 hours)**:

- There is a noticeable drop in consumption after the early morning peak, reaching its lowest point just before the morning hours. This suggests a period when the household is mostly inactive, likely because occupants are asleep.

ChatGPT 4

1. **Early Morning Peak (0-5 Hours)**: - The profile starts with a high peak early in the morning. This could indicate overnight usage of appliances such as heaters, especially if this is a winter day profile, or other devices left running through the night.

2. **Morning Drop (5-10 Hours)**: - After the initial peak, there is a significant drop in consumption. This decrease might correspond to a period when the residents are sleeping and most active appliances are turned off, or it could reflect a period just before the household starts its daily activities.

# 5. Conclusions

In this study, we explored the capabilities of advanced large language models (LLMs) like ChatGPT, Gemini, and Llama alongside the Retrieval Augmented Generation (RAG) approach using an electricity knowledge graph for answering complex questions about electricity datasets. Our analysis illustrated that while LLMs can provide quick and broadly accurate responses, integrating them with RAG, which pulls precise data from a specialized electricity knowledge graph, significantly enhances the precision and details available of the responses. This synergy between generative AI and targeted data retrieval proves especially beneficial in fields like energy data analysis, where precision and context-specificity are paramount. As such, RAG not only mitigates some common flaws in LLMs, such as the generation of plausible yet incorrect information, but also enriches the model's ability to handle specific, nuanced queries that are critical for data-driven decision-making in the energy sector. This integration represents a promising direction for further research and application, particularly in enhancing the reliability and utility of AI in specialized domains. However, a larger scale research effort on more domain specific data is needed to fully asses the robustness or RAG and the influence of potential build-in LLM bias that would disregard the context in the generated answer.

# 6. Acknowledgments

This work was funded in part by the Slovenian Research Agency under the grant P2-0016 and L2-50053.

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
