# OpenReview forum: "RAG Enabled Conversations about Household  Electricity Monitoring"
_KDD.org/2024/Workshop/DL4KG — Submitted to DL4KG 2024_

### Official Review · Reviewer_h4rr · 2024-07-02
**Lack of quantitative evaluation and proper evaluation metric reduced the generalizability of this work**

**Rating:** 2
**Confidence:** 3

**Review:**

# Summary
This paper presents an application of Retrieval Augmented Generation (RAG) in energy data analysis by integrating LLMs with a specialized electricity knowledge graph (KG). In order to verify the proposed approach, four (4) prompts were used to compare the answers of four (4) LLMs with and without the KG.

# Weak Points
-  The number of prompts used in the comparison is not substantial (only 4), which reduces the generalizability of the study and overestimates the applicability of RAG in this use case.
-  Quantitative evaluation is lacking.
-  No proper evaluation metric was used (i.e. Precision, Recall, F1-Score, etc.)
-  Description of how the natural language questions were converted to SPARQL queries are missing.
-  Figure 2 is not sufficient to describe the content of the KG used for evaluation.
-  Prompts: “Enumerate in one short sentence” - what is the purpose of the phrase/instruction “in one short sentence”? If one expects to get 20 distinct datasets, the expected result will not be a “short sentence”.

# Minor Comments:
-  Page 1, footnote: name of conference was not changed in the template.
-  Original RAG paper not cited (Lewis et al., 2020).
-  Page 2: “most in most of the cases”
-  Page 2: relational database… or a [triplestore] in case of a SPARQL query.
-  Page 2: last paragraph, repetitive text.
-  Figure 1: perhaps use a different term other than “transformer”, so as not to confuse with the transformer architecture.
-  Page 3: “can be queries via SPARQL”

# Detailed Score
## Clarity: 3 (fair)
*The writing is clear.*

## Quality: 2 (poor)
*See "Weak Points" above*

## Originality: 1 (questionable)
*The methodology is neither innovative nor a derivative. Novelty is lacking, other than using RAG approach to a specific use case."

## Relevance: 4 (clearly relevant)
*The work is relevant to the workshop as it tries to integrate a KG with LLMs."

## Impact: 1 (very low)
"See comment concerning evaluation."

---

### Official Review · Reviewer_1qLB · 2024-07-04

**Rating:** 4
**Confidence:** 5

**Review:**

The paper presents a qualitative analysis of how RAG can improve the performance of a LLM.

The authors designed 4 prompts, and tested them against 3 LLMs: gemini, chatgpt and llama.

The paper is well-written and relevant to the workshop's focus. However, it lacks novelty due to limited generalisability of the findings.

The main conclusion: "Our analysis illustrated that while LLMs can provide quick and broadly accurate responses, integrating them with RAG, which pulls precise data from a specialized electricity knowledge graph, significantly enhances the precision and details available of the responses." aligns with existing research and is somewhat expected..

Also, I am not sure the authors are exploring the full potential of RAG. For instance, in the "Prompt 1 for ChatGPT 4o with RAG" at page 4, the constructed prompt with RAG already includes all the sources (listed) necessary for the response, raising questions about the LLM's contribution.

Additionally, comparing LLMs vs LLM+RAG in this domain may not be entirely fair, as LLMs are inherently disadvantaged in a highly specialised knowledge area.

---

### Official Review · Reviewer_ihGH · 2024-07-07
**Review on RAG Enabled Conversations about Household Electricity Monitoring**

**Rating:** 4
**Confidence:** 4

**Review:**

This paper proposes to integrate Retrieval Augmented Generation (RAG) with large language models (LLMs) in order to improve information retrieval on electricity datasets. Three LLMs have been evaluated (ChatGPT, Gemini and Llama), overall improving the precision of answers.

The paper could gain in clarity if more details provided in Section 2 on the interactions between RAG and KG.  Section 3.1 also presents a set of prompts, without really discussing them. More importantly, the evaluation is limited to very few prompts (4), making hard to evaluate how the approach could generalise.

Minor comments:
- Ther answers to Prompt 1

---

### Decision · Program_Chairs · 2024-07-09

Reject